# Crowd Counting with Semantic Scene Segmentation in Helicopter Footage

**DOI:** 10.3390/s20174855

**Published:** 2020-08-27

**Authors:** Gergely Csönde, Yoshihide Sekimoto, Takehiro Kashiyama

**Affiliations:** 1Department of Civil Engineering, The University of Tokyo, 4-6-1 Komaba, Meguro, Tokyo 1538505, Japan; 2Institute of Industrial Science, The University of Tokyo, 4-6-1 Komaba, Meguro, Tokyo 1538505, Japan; sekimoto@iis.u-tokyo.ac.jp (Y.S.); ksym@iis.u-tokyo.ac.jp (T.K.)

**Keywords:** remote sensing, helicopter footage, deep learning, computer vision, image processing, crowd counting, semantic segmentation, multitask learning

## Abstract

Continually improving crowd counting neural networks have been developed in recent years. The accuracy of these networks has reached such high levels that further improvement is becoming very difficult. However, this high accuracy lacks deeper semantic information, such as social roles (e.g., student, company worker, or police officer) or location-based roles (e.g., pedestrian, tenant, or construction worker). Some of these can be learned from the same set of features as the human nature of an entity, whereas others require wider contextual information from the human surroundings. The primary end-goal of developing recognition software is to involve them in autonomous decision-making systems. Therefore, it must be foolproof, which is, it must have good semantic understanding of the input. In this study, we focus on counting pedestrians in helicopter footage and introduce a dataset created from helicopter videos for this purpose. We use semantic segmentation to extract the required additional contextual information from the surroundings of an entity. We demonstrate that it is possible to increase the pedestrian counting accuracy in this manner. Furthermore, we show that crowd counting and semantic segmentation can be simultaneously achieved, with comparable or even improved accuracy, by using the same crowd counting neural network for both tasks through hard parameter sharing. The presented method is generic and it can be applied to arbitrary crowd density estimation methods. A link to the dataset is available at the end of the paper.

## 1. Introduction

With the recent rapid developments in convolutional neural networks (CNNs), many image processing tasks that were very difficult a decade ago have become easier. Perhaps the most obvious example is the recognition of humanoid shapes in images. The detection and counting of humans in digital images, especially in crowded scenes, offer several applications, including traffic monitoring, safety surveillance, and finding stranded people following disasters.

At present, the task of image-based crowd counting can be divided into two main categories: direct and indirect methods. In the former case, all individuals are separately identified in the image, following which the total number of humans is obtained by counting those individuals directly. However, indirect methods produce additional abstract information, from which the total count can be achieved through a more elaborate procedure. In this study, we focus on indirect methods; specifically, density map estimator (DME) CNNs, although the work presented can also be applied to direct methods.

DME CNNs use an arbitrary three-channel RGB image as input and produce a single-channel crowd density map. Thereafter, the total count of humans can be obtained by integrating over the entire image. The advantage of this method is that, in addition to yielding the total count of humans, it provides a detailed map of their spatial distribution. The networks are usually fully convolutional, which means that the input can have an arbitrary size.

The typical footage type used for crowd counting is created with street-level cameras and drones flying at low altitudes. Although these image types are used extensively, they suffer from strong perspective distortions and obstructions, owing to the low camera angle. As a result, people appear in many different sizes and shapes in the imagery, and many methods can only detect a subset of these varieties, thereby yielding suboptimal density maps. Most works in the field are focusing on solving these issues at present.

Another issue exists that is unrelated to the previous problems and, to the best of the authors’ knowledge, is not currently being investigated by researchers. Although generic human detection has its uses, it also exhibits limitations from certain practical viewpoints. If the task is to identify a subset of humans, identifying that all humans will produce faulty results. The simplest example is the differentiation between actual living humans and fake ones, such as dummies, statues, or posters, which inevitably appear in urban footage whether or not they are exhibited in conventional datasets.

The set of features used for identifying humans can also be used for identifying human subcategories, depending on the task; for example, people with a certain hair color, wearing a certain type of clothing, or carrying a bag. However, more abstract subcategories exist that are difficult or impossible to identify based on humanoid features alone. One such subcategory is pedestrians.

Counting pedestrians offers several applications. Such data can be used to reconstruct the traffic flow of people, which can be subsequently used for further research purposes or monitoring the use of infrastructural objects. Safety monitoring is another useful application, and the monitoring of curfews is also an interesting option. Owing to the recent Coronavirus outbreak, many countries have imposed a certain level of restrictions relating to time spent on the streets or social distancing.

For the abovementioned purposes, people inside buildings, on balconies, or even on rooftops are irrelevant, as are people tending to their gardens or working at construction sites. Moreover, as mentioned previously, inanimate human-like objects may also exist, such as billboards and other forms of street advertisements (Figure 1). From the perspective of human detection, there is no real difference between an “image of a human” or an “image of an image of a human”. We must recognize the billboard itself to differentiate between the two. The latter is not specific to pedestrians, but it is a generic issue. Furthermore, seemingly random false positive detections may occur in the regions of an image where there are not supposed to be people. Finally, the density map may contain background noise, which distorts the accuracy.

Changing the footage type is an unorthodox means of solving the problems of perspective distortions and occlusions. As opposed to conventional datasets, helicopter footage is free of perspective distortions and it exhibits fewer obstructions. This is because of the high altitude and steep camera angle. With such a setup, there is a very small size difference between humans in different parts of the image, and even when they are standing close to one another, large parts of their bodies remain visible. Moreover, one image can cover a much larger area from a higher altitude than from the street level.

In summary, helicopter footage contains several advantageous attributes for human detection. Most importantly, it exhibits features that are not covered by other conventional datasets, as demonstrated in [1], in which a new helicopter dataset was introduced as part of the investigation. In this study, we introduce a new large-scale urban helicopter dataset, known as Tokyo Hawkeye 2020 (TH2020) [2], which shares the same attributes, but has a much larger image and pedestrian count, and it also exhibits inanimate human-like objects in the urban environment.

We propose a solution that also happens to help with false detections in regions where ordinary humans cannot possibly exist in order to address the issue of differentiating between all human-like entities and actual pedestrians. We explicitly apply semantic segmentation to enhance the DME results. The goal in the semantic segmentation task is to sort the regions of the input image into several semantic categories. If *C* different categories exist, the network generates *C* confidence score maps: one for every category. Semantic segmentation was also used as binary classification in several other works, which differentiated between human and non-human regions in order to attempt to enhance the accuracy. Our approach differs from this: in the footage that we introduce, the ratio of human regions is very low when compared to the background. Therefore, we ignore humans in the segmentation and focus on classifying the background, as illustrated in Figure 2.

Semantic segmentation for aerial footage has been well developed and it can achieve high accuracy. The segmentation can be used to identify the regions of images in which pedestrians can or cannot be found. We demonstrate that this approach can be used to improve the accuracy of pedestrian detection by masking regions where pedestrians cannot exist. Moreover, we show that the segmentation can be achieved by the same network as the density map estimation by performing optimization for a combinatorial loss (crowd density and segmentation) with minimal additional computational cost, while the density map accuracy does not decrease significantly or even improves. The former approach can be used with steep-angle, high-altitude footage, but is unfeasible with conventional crowd counting benchmark datasets. Although the latter approach would be applicable to arbitrary datasets, to the best of the authors’ knowledge, no other dataset has the required annotation information. Because the introduced methods are independent of the underlying DME network, they can be applied to any arbitrary architecture. To support this statement, we present experimental results using three different state-of-the-art DME networks.

## 2. Related Work

### 2.1. Crowd Counting

The majority of recent works have focused on solving two problems that arise in state-of-the-art crowd counting methods: varying scales and perspective distortions. Several researchers have diretly addressed the varying target sizes [3,4,5,6,7]. Others have experimented with increasing or changing the receptive field of the output pixels [8,9,10,11]. Certain scholars have attempted to tackle perspective distortions by using focused attention mechanisms [12,13,14,15,16]. Learning residual errors and correcting density estimations with these errors has also proven to be a viable approach [17,18,19].

All of these methods serve the purpose of improving the accuracy of generic human detection. However, we need to include the surroundings of the people in the calculations if we wish to differentiate between people on roofs, balconies, or posters and those on the street. Although certain methods may do so unintentionally and implicitly to an extent, our aim is to address this explicitly using semantic segmentation.

The use of semantic segmentation in crowd counting is not new [20,21,22,23,24]. All of these works aimed to create segmentation to separate human regions from non-human regions. In many situations, this is a simple binary classification, but even when multiclass segmentation is used, the problem is eventually reduced to human versus non-human separation.

In our work, we ignore humans as a segmentation class and focus on the segmentation of the background. We classify the background from the functional perspective of how likely it is that pedestrians exist in that region. This approach can be used with steep-angle, high-altitude footage, but it is unfeasible with conventional crowd counting benchmark datasets. We elaborate further on this in Section 5.1.

Besides density map methods, there is another closely related paradigm called localized regression. The key difference lies with the target map. In this approach, every map pixel aims to represent the total count in a small, localized region of the input. One typical way to create such a target map is to apply a uniform square-shaped kernel to the head annotations [25]. The result will be a map of redundant localized counts. Reference [26] also changes the counting task to a classification task by predicting count intervals instead of specific counts. These methods achieve competitive accuracy.

While most research in crowd counting is based on image processing techniques, it is worth mentioning that there are efforts underway to also use the rest of the electromagnetic spectrum for this purpose. Some methods rely on on-person devices, such as smart phones or radio frequency identification cards. The problem with these is that they are working with the assumption that people would carry the device at all times, not to mention the serious privacy issues that are raised by use of such devices. For indoor counting in buildings, where there are plenty of Wi-Fi enabled devices installed, personal device-free Wi-Fi based methods yield high count accuracy [27,28] while preserving privacy. However, in open outdoor spaces, these methods are less viable due to the significantly lower number of installed devices and the lack of refractive surfaces.

### 2.2. Semantic Segmentation

The first significant breakthrough in CNN-based semantic segmentation was the realization that the task is essentially no different from classification, and any arbitrary classification network can easily be converted into a segmentation network [29]. Subsequently, various network architectures were rapidly developed in an attempt to improve the segmentation accuracy. For example, FuseNet [30] incorporates depth information into the training in order to improve the segmentation accuracy.

An important issue in the segmentation task is that the network is usually required to provide a larger receptive field when compared to object detection. Dilated convolutions can solve this problem. DeepLabv3 [31] is a model based on ResNet [32], which uses dilated convolutions in its atrous spatial pyramid pooling layer.

Several architectures have been specifically developed for aerial semantic segmentation [33,34,35,36,37]. A common feature of the above methods is that they are multimodal or multispectral. The ISPRS aerial datasets [38,39] are typical benchmark datasets for aerial segmentation.

### 2.3. Existing Datasets

Numerous CNN-based human detection or counting methods have been developed and tested on standard datasets, such as the UCSD [40], INRIA [41], Caltech Benchmark [42], UCF_CC_50 [43], and ShanghaiTech [3] datasets. Although these image sets can be distinguished from one another by the average person count, they were all obtained from near-street-level cameras and from low angles. Therefore, these images contain many obstructions and perspective distortions. These issues pose substantially less difficulty in aerial footage.

Obstructions are caused by people walking alongside one another. In images that were taken from a steep angle, the obstruction is not very large because people do not walk over one another. Perspective distortion is caused by the relative difference between the distances of objects from the camera. The size of an object in an image is inversely proportional to its distance from the camera. Therefore, if the ratio of the distance of two objects is much larger (or smaller) than 1.0, the difference in their sizes in the image will also be very large. This ratio is closer to 1.0 in photographs taken from a high altitude. Figure 3 depicts these issues.

Of course, other aerial datasets are also available, which have been created either with airplanes or, more recently, using drones. Airplane footage is typically obtained from a high altitude, which is not suitable for the detection of small targets. The authors of [44] introduced a high-altitude, vertical-angle airplane dataset with acceptable resolution. The elevations are lower in the case of drone datasets, so human detection can be achieved. The angle is often vertical, which is advantageous for considering occlusion, but the human features are less recognizable. If the angle is low, perspective distortions occur, but a better viewpoint is provided for human features. Examples of drone datasets include the SDD [45], VisDrone2019 [46], and Okutama-Action [47] datasets.

Some of the above-mentioned datasets are image based, whereas others are obtained from videos. The image-based datasets were annotated manually. The common annotation strategy for video-based datasets is to annotate certain key frames, such as every tenth one, and then interpolate the annotation between frames. This vastly increases the dataset size, but not the unique object instance count. Certain video-based datasets, such as SDD and Okutama–Action, use unique ID-based annotation, so that it can also be considered for tracking.

## 3. Materials and Methods

### 3.1. Segmentation-Based Region Masking

During our preliminary experiments, we identified two main issues with pedestrian counting. The first is that even the best DMEs often produce false positive predictions and contain background noise in locations where humans cannot be present, such as flat wall surfaces or treetops. The other is that, even though the annotation only contains pedestrians, the trained models learn to identify humans regardless of the context and, as a result, non-pedestrians and poster images of humans are also identified and counted.

Both of the above issues can be solved by identifying the regions in which pedestrians cannot be present. We refer to these as invalid regions. Consequently, we refer to regions in which pedestrians may be present as valid regions. The invalid region information can be used to modify a base crowd density map. We do so by integrating the density map estimation and semantic segmentation. We use the mean absolute error (MAE) primarily and root mean squared error (RMSE) secondarily for the density map quality measurement metrics.

In the following sections, we explain two methods for region-based masking, but, first, we introduce our dataset in detail because it is essential for our methods.

### 3.2. Dataset

We introduce a new helicopter-based pedestrian dataset. There are two main reasons for this. Firstly, a dataset from such a viewpoint exhibits features that are not covered by other conventional datasets, and models trained on those datasets show very weak performance on helicopter footage, as demonstrated in [1]. Secondly, the method that we introduce depends on the attributes of the dataset.

Because our dataset is an extended version of the TH2019 dataset [1], we named it TH2020. It has the same attributes as TH2019, except for the dataset size; there are roughly 20 times more images and 15 times more pedestrian annotations in our dataset. Specifically, TH2020 comprises 6237 static images with a resolution of 960×540, containing 120,772 pedestrian annotations from 10 different locations and 22 different sessions. The images were obtained from helicopter footage from a high altitude at a steep, but not vertical angle. Owing to the high altitude, the perspective distortions and scale differences are negligible, although the average target size is very small. It is important to note that, because the camera angle is not vertical, pedestrians are partially visible from the sides, thereby increasing the number of recognizable features as compared to those of a vertical camera angle, which is the most typical in airplane footage. Figure 4 presents some sample images.

The dataset includes two types of ground-truth annotations: a pedestrian annotation and a semantic segmentation annotation. The former only contains pedestrian head coordinates, including bikers, as there is no reason to differentiate between these from a practical viewpoint. Moreover, distinguishing bikers from pedestrians is very challenging; for example, if there are 10 people surrounding a bicycle, any one of these may be the rider. People on rooftops or on construction sites are excluded from the ground-truth. We generated a ground-truth density map from the head coordinate annotation using the conventional method. We created a matrix with the same dimensions as the input image. The matrix contained 1 s at the head coordinates, and 0 s otherwise. Thereafter, we convolved this matrix with a Gaussian kernel. We used a fixed kernel with a standard deviation of 15 instead of the adaptive method.

For the segmentation, only 5040 images were annotated, owing to financial and time limitations. The annotation assigned one category to every pixel of the image. When we created the segmentation ground-truth, we were not aiming for an extensive annotation. Rather, we attempted to achieve simplicity and focused on a functional perspective for pedestrian detection. We wanted to ignore objects that were too small to occlude humans completely, so we considered those objects as part of their surrounding entities. Moreover, we required high-level classes, as we did not want the annotation to be too detailed. Therefore, we decided to annotate the following categories:Road: areas designated primarily for vehicle traffic; humans are often present.Sidewalk: areas designated for pedestrian traffic; humans are often present.Building: areas covered by construction that pedestrians can enter (houses, offices, etc.); humans may appear in windows, on balconies, and on rooftops.Structure: areas covered by construction, often with girder-like features, where entry is not possible (walls, columns, railings, etc.); humans are unlikely.Plant: areas covered by vegetation that can block visibility, such as trees with leaves and large bushes; humans are unlikely.Vehicle: areas covered by cars, buses, trucks, etc.; depending on the vehicle and lighting conditions, human heads may be present, but they are unlikely.Background: anything else; typically grass, rubble, railway tracks, etc.; humans may be present, but are unlikely.

With regard to privacy matters, it should be mentioned that the resolution of the footage is not enough to identify any individual; only coarse features, such as hair, clothing, or luggage, can be identified. Therefore, there are no privacy issues with the dataset, as opposed to conventional benchmark datasets, where there are clearly recognizable people.

We divided the dataset into training and evaluation parts scene-wise instead of applying a random split in order to avoid overfitting. The training set was the same for the segmentation and density map estimation, whereas the evaluation set for the segmentation was a subset of the density map estimation evaluation set, as not all images had segmentation ground-truth. Specifically, there were 4114 training images, 926 segmentation evaluation images, and 2123 density map estimation evaluation images.

### 3.3. Simple Masking with Separate CNNs

In this approach, we used an arbitrary DME CNN and an arbitrary semantic segmentation CNN, and trained both on the TH2020 dataset. For inference, we used the input image and generated an ordinary crowd density map with the unaltered DME CNN. Thereafter, we used the semantic segmentation CNN to generate a segmentation map for the same image. This segmentation map could have arbitrary classes. In our experiments, we used those that were explained in Section 3.2. Subsequently, we took the segmentation map and simplified it as a valid–invalid region map based on a predetermined table. Valid pixels had a value of 1, whereas invalid ones were 0. As the two CNNs were independent from one another, their output resolutions could differ. In such a case, we resampled the valid–invalid map to the DME output resolution using nearest-neighbor interpolation. As the final step, we masked the original crowd density map by multiplying it element-wise with the valid–invalid map. Alternatively, it would be possible to assign real numbers between 0 and 1 to every segmentation class and to use that as a mask, but, because this was our first attempt with masking, we aimed for simplicity. Therefore, we used the zero–one map. Figure 5 illustrates the flow.

There is more than one means of creating a valid–invalid map from segmentation. The most obvious method is that, whereby for every pixel, the value for that segmentation class from the predetermined table is inserted. Although this is certainly a viable approach, it has two drawbacks. Firstly, the segmentation is not perfect, so it is possible that some estimated invalid regions will overlap with some pedestrians. Secondly, it is very common for pedestrians to appear at the edges of invalid regions because the camera angle is not vertical. For example, when someone walks in front of a building and very close to it, most of the body will overlap with the building.

We proposed max-pool masking to solve the above problems. We applied a max-pool layer to the valid–invalid map with a carefully selected kernel size (we used a kernel size of seven) and a single pixel stride. That is, an invalid pixel would only remain invalid if it did not have any valid pixels in a certain vicinity. In this manner, the sizes of the invalid regions were shrunk to counter the effects of segmentation errors and natural overlaps.

### 3.4. Dme Segmentation Multitask Training

Any arbitrary classification network can easily be converted into a segmentation network, as demonstrated in [29]. However, we decided not to limit our approach to classification, as many image processing CNNs are based on classification networks in any case. Furthermore, technically, any arbitrary fully CNN can be changed into a semantic segmentation network by replacing the final output layers and loss function, which is the approach we used.

We used crowd counting CNNs, increased the number of channels in the final output layer to C, and changed the loss function to a conventional segmentation loss, specifically, the softmax cross-entropy loss. However, it occurred to us that the output did not contain any parameters, which meant that the model was exactly the same as that used for crowd counting, so we could attempt to perform the two tasks simultaneously. Moreover, we realized that, because we were generating a crowd density map and segmentation map at the same time, we could use the segmentation map directly for masking the density map. Thus, our final architecture contained C + 2 output channels: one for crowd density map estimation, C for the segmentation categories, and one for the masked crowd density map. Figure 6 presents a graphical layout of our proposed method.

For the loss function, we calculated the Euclidean norm over the density map per-pixel errors and softmax cross-entropy for the segmentation channels, and combined these two losses with a balancing factor α.
(1)$$L_{c}=\frac{1}{N}\sum_{i=0}^{N}{\left(d_{i}-{\hat{d}}_{i}\right)}^{2},$$
(2)$$L_{s}=\frac{1}{N}\sum_{i=0}^{N}-log\left(\frac{e^{-p_{i}^{g}}}{\sum_{c=0}^{C}e^{-p_{i}^{c}}}\right),$$
(3)$$L=L_{c}+\alpha L_{s},$$
where *N* is the number of pixels in the mini-batch, $d_{i}$ and $\hat{d_{i}}$ are the ground-truth and estimated per-pixel crowd density values, respectively, *C* is the number of categories, $p_{i}^{g}$ is the predicted confidence score for the ground-truth category at pixel *i*, and $p_{i}^{c}$ is the predicted confidence score for category *c* at pixel *i*.

The Euclidean norm could also be calculated over the masked crowd density map instead of the raw one, but, in this case, finding the correct alpha was very difficult, because the valid–invalid map could easily become stuck in the complete 0 or complete 1 state.

In this approach, we also evaluated the semantic segmentation quality, for which we used the mean intersection-over-union (mIoU).

We added the MTSM prefix (for MultiTask Segmentation Method) to distinguish between the original backbone crowd counting network and the multitask version. For example, if the backbone was CSRNet architecture, we referred to our modified version as MTSM-CSRNet.

Finally, it should be noted that the conversion method also works backwards; that is, a semantic segmentation network can be converted into a DME network, but we have not conducted such experiments yet and, thus, the accuracy of the backwards direction remains to be determined.

## 4. Results

### 4.1. Simple DME and Segmentation Methods

In our experiments, we used CSRNet [9], CAN [48], and SPNet [10] as the DME networks and DeepLabv3 as the semantic segmentation network, although, as we will demonstrate, there is no real difference. We selected a general-purpose segmentation network, because methods for aerial footage also require height information, which we did not have for our footage.

As we were using a custom dataset instead of a conventional benchmark dataset, we first had to establish a baseline accuracy value for all of the networks. The DME results can be viewed in the first line of Table 1, whereas Table 2 and Table 3 display the DeepLabv3 results. In the segmentation task, we assigned valid values to the road, sidewalk, and background classes. The average pedestrian count in the evaluation set was approximately 30, which means that the relative MAE was slightly over 0.2. We were confident that the models were well trained, as these models produced similar relative errors on conventional datasets. In the case of DeepLabv3, the evaluation was not so simple. We could not compare the results to those of aerial segmentation datasets, as those also contain height information, which aids in achieving very high mIoU values. Our best option was a comparison with the Cityscapes dataset [49]. Deeplabv3 achieved an mIoU of 81.3 on Cityscapes. The accuracy achieved on our dataset was slightly worse than this, but the annotation was very different, so this difference did not mean that the trained model was not as effective as it could possibly be.

Moreover, we made several attempts with SANet [50], because the authors claimed to achieve high accuracy on conventional benchmark datasets, but we could not train the model. Even the lowest error we could achieve was almost three times higher than that of the other networks. Therefore, we excluded it from our investigation.

### 4.2. Simple Masking with Separate CNNs

We took our most accurate segmentation model, which happened to be MTSM-CAN, and used it to mask the density maps for all three DME networks. The MAE improved in all scenarios. The results are displayed in the second line of Table 1. Figure 7 presents a comparison of the different models.

### 4.3. DME Segmentation Multitask Training

For this method, we modified CSRNet, CAN, and SPNet according to Section 3.4. The DME results are displayed in the third and fourth lines of Table 1.

We reported the results for the models with the lowest MAE and reasonable segmentation quality. The latter means that the softmax cross-entropy loss was lower than 0.2. Table 2 and Table 3 present the per-class and valid–invalid IoUs as compared to the same metrics of DeepLabv3. Figure 7 displays a comparison among the models.

The output of DeepLabv3 had the same resolution as the original image, whereas our backbone networks applied downsampling. To enable a fair comparison, we reported the mIoU for both the downsampled segmentation map and the downsampled segmentation map upsampled back to the original resolution.

CSRNet and CAN were demonstrated to be strongly adaptable to our MTSM model, with comparable and even improved accuracy. In the case of SPNet, both the counting and segmentation metrics decreased. The most important difference between SPNet and the other architectures is that SPNet is shallow when compared to the others.

It can be observed from Table 1 that the masking actually decreased the counting accuracy. In the following section, we investigate the reason for the varying masking efficiency.

### 4.4. Masking Efficiency

The level of improvement depends on the quality of both the density map estimation and semantic segmentation. The dependence on the segmentation quality is straightforward. If the invalid areas are larger in the estimation than in the ground-truth map, true positive detections can be masked out. However, nothing may change if they are smaller than the ground-truth.

The dependence on the crowd density estimation quality is slightly more complicated. For argument’s sake, let us assume that the segmentation map is correct. This is a safe assumption, because we achieved very high invalid region IoUs (Table 3), and the possibility of errors is further reduced by the max-pool masking. Moreover, let us assume that the density map does not contain negative elements. Although many architectures do not enforce this, in our experiments the robust networks eliminated negative values (or reduced them to very close to 0).

If the total count is overestimated, then there must be several false positive detections. As we assumed perfect segmentation, the masking can only remove false positives. If there is a removed false positive count that is more than twice the original error, the MAE will increase. However, this also means that the estimation has numerous false negatives and the network attempts to compensate for the error with false positives.

If the total count is underestimated and something is still masked out, thereby increasing the MAE, the model produces false negatives and false positives simultaneously, but the false negatives outweigh the false positives. In summary, the masking efficiency depends on whether or not the estimated density map is littered with both false positives and false negatives.

If there are many false positives or false negatives, then they will cause spike-like errors in the density map. These spikes significantly increase the L2 norm of the pixel errors (that is, the Euclidean loss). In contrast, if there is mostly only background noise, the L2 norm will be much smaller than the L1 norm. Therefore, the ratio of the two norms provides information regarding the amount of spike-like errors; that is, the number of false positives and false negatives in the estimation. We collected this information and plotted it against the MAE improvement, which can be observed from Table 4 and Figure 8.

### 4.5. Training Details

We downloaded open-source implementations for all models except for SPNet, which we implemented ourselves (excluding the final ReLU layer).

As DME networks are downscaling and sometimes upscaling, the output resolution differs from the original resolution. Thus, because the ground-truth has the same resolution as the original image, it must be resampled. We used bicubic interpolation for the density map and nearest-neighbor interpolation for the segmentation map. For the density map, we also had to multiply the pixel magnitudes by the resampling factor in order to preserve the total count. We applied a random horizontal flip as augmentation. As the footage had a well-defined upwards vertical direction, it was pointless to apply rotation or a vertical flip, and we did not want to complicate the training further. As all three DME networks have a VGG-16 [51] backbone, we initialized them using pre-trained VGG-16 parameter weights. There were no suitable pre-trained weights available for DeepLabv3; therefore, we had to train it from scratch. The parameters without pre-trained values were randomly initialized with a normal distribution (μ=1,σ=0.02).

We trained the models on full images, because DeepLabv3 struggled with patch-wise training, and it was important for all of the models to be trained on the same dataset for comparison. We used one or four GPUs for the DME methods. In the case of four GPUs, the batch size was increased, but we did not observe any change in the accuracy, only in the execution speed. We used four GPUs for DeepLabv3 owing to the large memory requirement. Each GPU had 16 GB memory in which a batch of two images could fit, yielding a total batch size of 8. For the DME, we set the batch size to range from 12 to 16 per GPU (the maximum number that could fit). We used the Adam optimizer with an initial learning rate of $10^{-4}$ and a momentum of 0.95. The learning rate was directly halved every 20 epochs, starting from the 40th epoch.

The details of the MTSM training were mostly the same as those in the case of the baseline models. The baseline model parameters were used as initial values for MTSM-CSRNet and MTSM-CAN. As MTSM-SPNet did not perform effectively when initialized from the baseline model, we launched several training sessions from scratch in order to achieve the highest accuracy.

However, the most important aspect was to determine the appropriate value for α. The Euclidean loss and softmax cross-entropy are very different functions. Therefore, using a simple constant scaling factor would result in the gradient components having different weightings in different sections of the training. During our training attempts on MTSM-CSRNet and MTSM-CAN, we often observed that with a constant α, the training shifted towards either the crowd density or segmentation very rapidly. We used an initial α of $10^{-6}$ and, whenever the training became too one-sided, we shifted it towards the weaker loss, which usually meant increasing α. In the case of MTSM-SPNet, the training could be performed by calculating the loss function over the masked density map using a constant α of $10^{-5}$. In practice, we used two balancing factors: one for each loss component, so the final loss would be closer to 1, because small fractions are difficult to read.

## 5. Discussion

### 5.1. Correlation between Density Map Quality and Masking Efficiency

Our data exhibited a linear correlation between the average L2/L1 norm ratio of the images and the change in the MAE caused by masking. This means that a model with a smaller average L2/L1 norm ratio can benefit more from the masking method. A factor that could not be observed from our experimental results is that models with very different MAEs may have the same average L2/L1 norm ratio. For example, if there are two models and one produces proportionally larger per-pixel errors than the other, then it will also have proportionally larger MAEs, while the average L2/L1 norm ratio will be the same. However, if the per-pixel errors are all proportionally larger, the per-pixel errors that are masked will also be proportionally larger, which is exactly the change in the MAE. In summary, based on the MAE value, the same average L2/L1 norm ratio may be correlated to different changes in the MAE. This suggests that the average L2/L1 norm ratio is probably linearly correlated to the change in the MAE relative to the MAE, but, as our experimental results had MAEs that were very close to one another, there was also a correlation with the absolute change.

Referring back to the results in Table 1, it can be observed that, although the MAE values improved, the RMSE increased slightly. This may occur if there are moderate improvements in images with small errors along with small degradations in images with large errors. If the dataset has a small variance in the per-image ground-truth count, a high RMSE indicates that images exist in the dataset that are greatly over- or underestimated. However, if the dataset has a large variance in the per-image ground-truth count, a high RMSE is inevitable, because every method exhibits a certain proportionality between the ground-truth count and absolute error; that is, if the ground-truth has high variance, the absolute error will also have high variance. This means that the RMSE is not a very useful metric for datasets with high ground-truth variation, such as ours, unless a method is developed that produces an error that is independent from the ground-truth count.

Moreover, it should be noted that the proposed masking method is not feasible with conventional crowd counting benchmark datasets for two reasons. Firstly, in conventional datasets, the ratio of the area that is covered by humans is very large, and often almost the entire image is covered with humans. Therefore, they cannot be ignored in the segmentation. Secondly, even if segmentation could be achieved by ignoring the humans, owing to the low camera angle and close distance, the heads would overlap excessively with invalid regions and, therefore, they would be masked out.

### 5.2. MTSM Model Parameters

Our aim was to delve deeper into the inner workings of the MTSM models. We were interested in understanding how these are realized, and how they differ from training two independent networks for crowd counting and segmentation apart from the obvious memory and/or computational requirements. We were dealing with two different tasks that could either be intertwined or not. The latter case means that inside the network, two parallel subnetworks are realized and the channels from the one do not affect the output of the other. However, if the two tasks are intertwined, there cannot be such separation and the parameters affect both outputs.

The existence of two separable subnetworks would mean that the current architectures for crowd counting are excessive and the same accuracy can be achieved with a substantially smaller model. However, our experiments demonstrated that this is not the case. If there were two subnetworks, the parameters in the final layer would have to be separable into two groups by affecting either one output or the other, which we investigated. For every output channel in the final layer, we calculated the absolute mean of its parameters over the input channels. Thereafter, we verified whether there were any input channels with parameters that were thousands of times smaller than the mean. If so, the given output channel would barely be affected by those input channels.

We found that only a handful of input channels exhibit such behavior, and most of these do not affect any output channels, regardless of the type. Therefore, we can confirm that the two tasks are strongly intertwined. In fact, there is no difference between them in the sense that both tasks are performed using the same feature set, and they are only divided by the form of the output. Our point is that well-designed architecture performs effectively in more than one image processing task, and not just in a very particular narrow field. Therefore, combining the task of crowd counting with other computer vision tasks to save on computation time offers development potential and future research possibilities.

Furthermore, whereas region-based masking is unfeasible for different types of datasets, the multitask segmentation method can be applied to any human dataset. Unfortunately, to the best of our knowledge, the TH2020 dataset is the only one that includes both human head annotation ground-truth and semantic segmentation ground-truth; therefore, we cannot conduct experiments on any other datasets.

### 5.3. Considerations about the Dataset

We designate the dataset as helicopter footage; however, the attributes of the dataset are what more significant. The most important of these is the high altitude. High enough so that perspective distortions and size differences can be markedly reduced with a steep camera angle while still covering large areas, but not so high that the targets become too small for detection. While technologically it is feasible to create such footage with drones, there are a few reasons why the use of helicopters is the only option currently.

In most countries, the use of airspace is very strictly regulated. Moreover, in many large metropolitan regions, commercial or public use of drones is prohibited. In regions with more flexible regulations, the use of drones is tied to special permissions but only allowed up to a very limited altitude. Certainly, if authorities are considering putting aerial surveillance systems in place, they may change these regulations for cost efficiency. However, it is still important to keep in mind technological requirements. Due to the high altitude, the camera equipment must have high zooming capabilities to achieve sufficient resolution. Moreover, precise aiming equipment is required to find and keep the target region in focus. These devices greatly increase the payload weight, thereby requiring larger drones with larger energy consumption and shorter flight times.

We also want to mention that there are practical applications where the use of drones is not possible at the current technological level. One such example is a search and rescue operation, where simultaneous crowd counting and semantic segmentation is a helpful tool to identify people in immediate danger. For the rescue, helicopter is the only option and splitting the operation into search and rescue parts to save money may cost time and, consequently, human lives.

## 6. Conclusions

We have introduced a new large-scale human dataset obtained from a helicopter. Based on the experimental results, the dataset itself is not particularly challenging owing to the lack of scale differences and perspective distortions. However, it offers a viewpoint that is not covered by other conventional crowd counting benchmark datasets. Moreover, the area covered by one image is much larger than that of conventional crowd counting datasets, meaning that the same region requires less computational resources to process.

The focus of our work was DME crowd counting. We have introduced a new method of extracting deeper semantic information regarding humans. Specifically, we have changed the task from crowd counting to pedestrian counting, which cannot be performed simply by detecting human features. The generated density map must use some information from the surroundings of the humans. We achieved this by applying semantic segmentation to the background, ignoring humans, and using these results to mask regions of the crowd density map where the segmentation indicates a non-pedestrian area.

Our experiments demonstrated that if the segmentation quality is high, the efficiency of this masking method is dependent on the quality of the density map estimation. In fact, the change in the accuracy exhibits a strong correlation with the ratio of the L2 norm and L1 norm of the per-pixel estimation error. In particular, estimations with lower L2/L1 ratios (that is, better estimation quality) benefit more from the masking method, whereas estimations with a high L2/L1 ratio may exhibit a decrease in accuracy.

We have also theorized that the density map and segmentation map can both be generated by the same network, resulting in a minimal increase in the memory and computational requirements compared to either task alone. We tested this theory with three different backbone networks, among which the two deeper architectures exhibited comparable or even improved accuracy. This means that there is no real difference between the image processing tasks in the sense that both tasks are performed using the same feature set, and well-designed models can perform effectively in a wide variety of problems. Therefore, combining crowd counting with other computer vision tasks is a valid research direction.

We attempted to replicate the work of other researchers appropriately where necessary, and to confirm the validity of open-source software resources. Furthermore, we executed our experiments with the utmost care to avoid any hazards that could falsify our results.

## Figures and Tables

**Figure 1 sensors-20-04855-f001:**
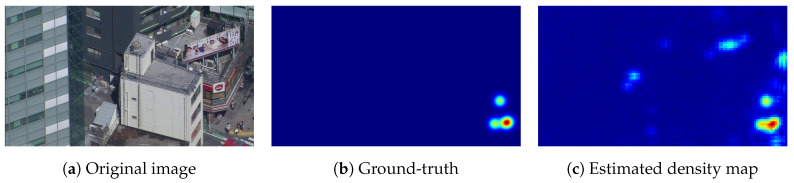
Sample image exhibiting the problem of “fake human” detection. It is clear that the billboard with humans on top of the building produces false positive detections.

**Figure 2 sensors-20-04855-f002:**
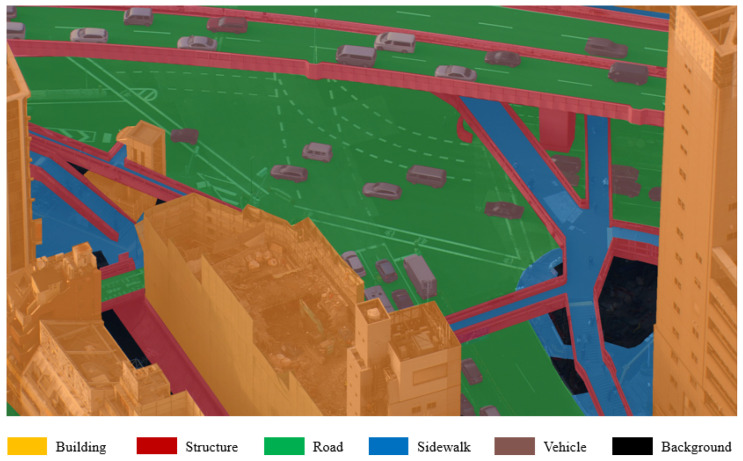
Semantic segmentation of background for helicopter footage, where humans are ignored.

**Figure 3 sensors-20-04855-f003:**
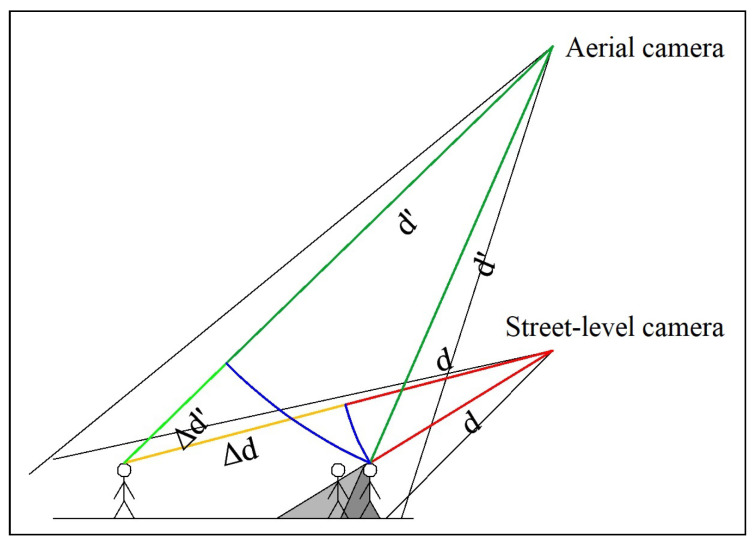
Illustration of differences between street-level and aerial footage. (d+Δd)/d is much further from 1.0 than $(d^{\prime }+\Delta d^{\prime })/d^{\prime }$. Moreover, the dark gray area is obstructed from both cameras, whereas the light gray area is only obstructed from the street-level camera. Note that the image is not to scale.

**Figure 4 sensors-20-04855-f004:**
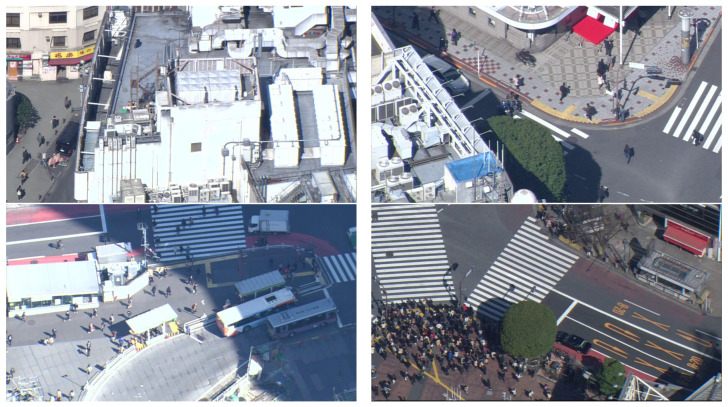
Sample images from TH2020 dataset.

**Figure 5 sensors-20-04855-f005:**
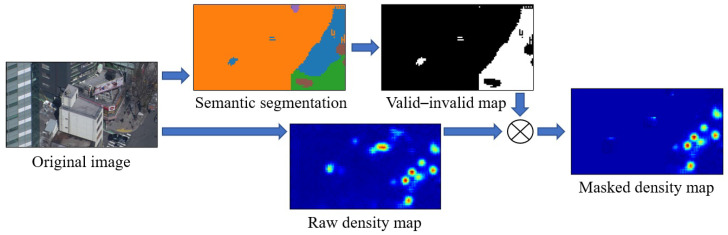
Flow of region-based masking procedure.

**Figure 6 sensors-20-04855-f006:**
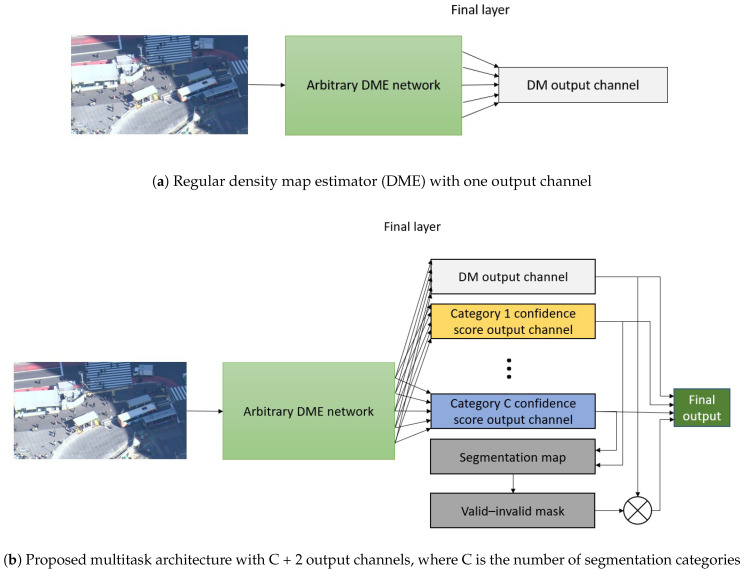
Visual representation of difference between regular DME network and our proposed multitask segmentation method.

**Figure 7 sensors-20-04855-f007:**
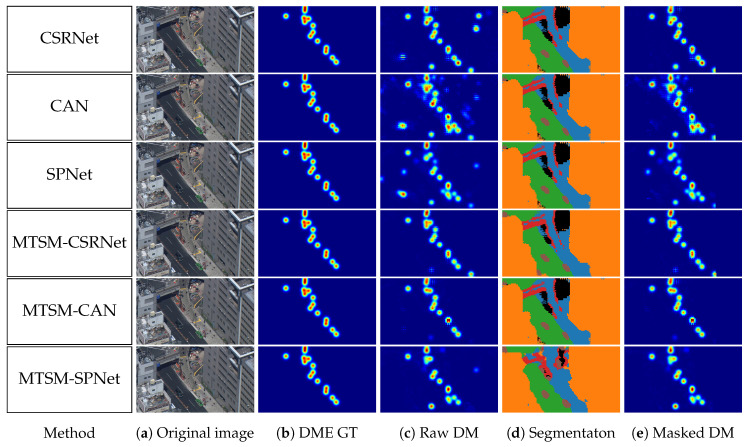
Sample results for comparison across methods. The segmentation for the regular density map (DM) methods was generated by MTSM-CAN.

**Figure 8 sensors-20-04855-f008:**
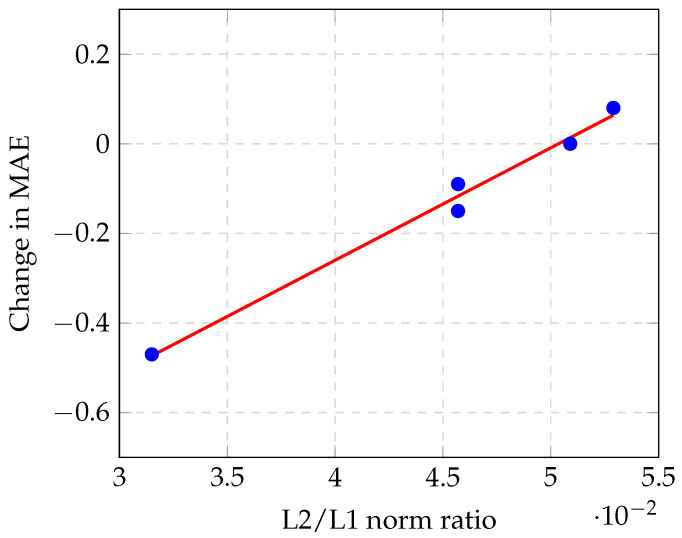
Correlation between DME quality and masking efficiency in case of high-quality segmentation mask.

**Table 1 sensors-20-04855-t001:** Summary of crowd density estimation accuracies for models trained on TH2020. The mean intersection-over-union (mIoU) column is only included to indicate the accuracy of the segmentation mask used.

Method		CSRNet			CAN			SPNet		
		MAE	RMSE	mIoU	MAE	RMSE	mIoU	MAE	RMSE	mIoU
Simple DME		7.35	19.63		6.5	13.54		6.29	16.97	
Masked	Simple DME	7.26	19.9	0.7562	6.03	13.95	0.7562	6.14	17.15	0.7562
	MTSM	6.84	20.94	0.7332	6.9	22.3	0.7562	6.58	19.56	0.5821
Raw		6.84	20.85		6.82	22.2		6.56	19.56	

**Table 2 sensors-20-04855-t002:** Per-category intersection-over-unions (IoUs) and means for DeepLabv3 and our MTSM models trained on TH2020.

Network	Building	Structure	Plant	Sidewalk	Vehicle	Road	Background	Mean	Upsampled Mean
DeepLabV3	0.8494	0.3431	0.6552	0.7905	0.732	0.8632	0.6765	0.7014	0.7014
MTSM-CSRNet	0.9207	0.6263	0.6115	0.7571	0.7004	0.8584	0.6583	0.7332	0.7131
MTSM-CAN	0.9428	0.6693	0.642	0.7632	0.7072	0.8834	0.6858	0.7562	0.7358
MTSM-SPNet	0.9043	0.4162	0.4541	0.673	0.4674	0.8308	0.3288	0.5821	0.5778

**Table 3 sensors-20-04855-t003:** Valid–invalid IoUs and their means for DeepLabv3 and our MTSM models trained on TH2020.

Network	Invalid (Building-Structure-Plant-Vehicle)	Valid (Road-Sidewalk-Background)	Mean
DeepLabV3	0.8916	0.8846	0.8881
MTSM-CSRNet	0.8999	0.8979	0.8989
MTSM-CAN	0.9112	0.9115	0.9113
MTSM-SPNet	0.8533	0.8508	0.852

**Table 4 sensors-20-04855-t004:** L2/L1 norm ratio as compared to change in accuracy after masking for investigated models.

Method	Average L2/L1 Norm Ratio	Change in MAE	Invalid Mask mIoU
CSRNet	0.0457	−0.09	0.9112
CAN	0.0315	−0.47	0.9112
SPNet	0.0457	−0.15	0.9112
MTSM-CSRNet	0.0509	0	0.8999
MTSM-CAN	0.0529	0.08	0.9112
MTSM-SPNet	0.0444	0.02	0.8533

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
