# Peer review of "Crowd Counting with Semantic Scene Segmentation in Helicopter Footage"

_sensors, 2020, doi:10.3390/s20174855_

Round 1
Reviewer 1 Report
The research presented in the paper presents a idea with good relevance in image based crowd monitoring. The presentation is almost flawless and easily understandable. The fact that the authors share their dataset publically, will be of definite help to the researchers in this domain. Although it is totally understandable that the authors want to focus on vision based crowd monitoring techniques, there can be some questions raised regarding on the introduction part.
Following are some comments, which can be addressed in the manuscript to make it more relevant:
- Please comment on usability of helpicopter based monitoring. (A lot has been said in the manuscript already). But, in my opinion, modern drones are capable of reaching sufficient heights to get the same angle of view as seen in the images presented in paper. Drones might be sufficient and more economic.
- Authors rightly pointed out the accuracy being almost improved to the point of saturation in crowd counting. Semantics is (not new in crowd vision) but still very important in modern AI applications. I believe, in the introduction or abstract authors must empahsize on a larger picture rather than just accuracy. Having semantic/ situational awareness will not only improve accuracy, but help in explainability, verification of results, preventing false results (already discussed a bit in manuscript).
- Although the focus is on vision based techniques, it is important to note that there is a surge and definitely a shift in non-vision based techniques. In view of this, authors must discuss through a 1 or 2 paragraph intro, the pros and cons of their technique as compared to WiFi, Radar, EM signature based crowd monitoring/ counting.
- Privacy is also important to discuss, specially when compared with non-vision techniques, which preserve privacy to a greater extent as compared to cameras. And how good of a choice is the technique presented in this paper, when it comes to data privacy?
Author Response
"Please see the attachment."

Reviewer 2 Report
This paper describes a new crowd counting dataset from the helicopter view. This dataset TH2020 extends their previous version TH2019 with significantly more images (I think a table comparing the difference between TH2019 and TH2020 may help). Different from standard crowd counting datasets that only have dotted annotations, the proposed dataset provides multi-class segmentation masks, which also enables simultaneous evaluation of counting and segmention and how each other affects counting. In this respect, this dataset is a valuable contribution to the community. The authors also provide a comprehensive benchmark on their dataset with thorough discussions of the results. In addition, I find the paper is very well written.
I only have a concern in related work. The authors also discussed about density-map-based deep counting networks. However, there is also another closely related counting paradigm called local count networks. This branch of work include:
- Count-ception: Counting by fully convolutional redundant counting, ICCVW, 2017
- Counting everyday objects in everyday scenes, CVPR, 2017
- TasselNetv2: in-field counting of wheat spikes with context-augmented local regression networks, Plant Methods, 2019
- Counting objects by blockwise classification, TCSVT, 2019
- From open set to closed set: Counting objects by spatial divide and conquer, ICCV, 2019
- Adaptive Mixture Regression Network with Local Counting Map for Crowd Counting, ECCV, 2020
Particularly, in 8th reference, it has demonstrated that local count regression provides a tigher surrogate than density map regression in theory. I expect to see a review of this branch of works for crowd counting.
Author Response
"Please see the attachment."
